# Dimensionality Reduction and Classification of Hyperspectral Remote Sensing Image Feature Extraction

**Hongda Li, Jian Cui, Xinle Zhang, Yongqi Han and Liying Cao ***

College of Information and Technology, Jilin Agricultural University, Changchun 130118, China
* Correspondence: caoliying@jlau.edu.cn

**Abstract:** Terrain classification is an important research direction in the field of remote sensing. Hyperspectral remote sensing image data contain a large amount of rich ground object information. However, such data have the characteristics of high spatial dimensions of features, strong data correlation, high data redundancy, and long operation time, which lead to difficulty in image data classification. A data dimensionality reduction algorithm can transform the data into low-dimensional data with strong features and then classify the dimensionally reduced data. However, most classification methods cannot effectively extract dimensionality-reduced data features. In this paper, different dimensionality reduction and machine learning supervised classification algorithms are explored to determine a suitable combination method of dimensionality reduction and classification for hyperspectral images. Soft and hard classification methods are adopted to achieve the classification of pixels according to diversity. The results show that the data after dimensionality reduction retain the data features with high overall feature correlation, and the data dimension is drastically reduced. The dimensionality reduction method of unified manifold approximation and projection and the classification method of support vector machine achieve the best terrain classification with 99.57% classification accuracy. High-precision fitting of neural networks for soft classification of hyperspectral images with a model fitting correlation coefficient ($R^2$) of up to 0.979 solves the problem of mixed pixel decomposition.

**Keywords:** hyperspectral images; data dimensionality reduction; feature extraction; machine learning; cell decomposition

## 1. Introduction

Since the 20th century, the widespread application of remote sensing technology reflects various surface information such as that related to agriculture, forestry, water, soil, minerals, energy, and oceans [1–3]. Hyperspectral remote sensing imaging promotes the development of precision agriculture [4]. Hyperspectral remote sensing images usually contain hundreds of bands with small wavelength intervals between each band. These images provide a large amount of extremely rich data for the target area [5], as well as facilitate considerable finer ground object information [6]. However, in some specific practical applications, a higher amount of data does not necessarily result in an increased amount of information because excessive information redundancy exists in hyperspectral images [7]. To extract all characteristic terrain data from hyperspectral data, dimensionality reduction must be performed according to data characteristics and band types. Hyperspectral image downscaling methods focus on feature extraction based on the linear and nonlinear, supervised and unsupervised, spatial and spectral features of the images. Hyperspectral images are processed by downscaling and classification. Deep learning has been a powerful feature extraction tool widely used for hyperspectral image classification, and convolutional neural networks (CNN) are capable of extracting nonlinear features [8,9].

Current hyperspectral dimensionality reduction methods are classified into feature extraction [10] and band selection [11]. Hyperspectral feature extraction refers to the

recombination and optimization of the original spectral–spatial features to extract new features that are most suitable for the application requirements. Feature extraction is usually performed in terms of feature vectors [12] and feature kernels [13] of hyperspectral data for dimensionality reduction. Dimensionality reduction attempts to retain local and global features. The global feature transforms the entire remote sensing data, and the local feature extracts the part of the data with obvious data features. Many methods of hyperspectral image data classification have been developed; for example, hyperspectral data classification based on deep learning [14], hyperspectral remote sensing image classification using the support vector machine (SVM) [15], semi-supervised learning image classification [16], and dimensionality reduction methods for image classification [17]. The most popular application is deep learning for remote sensing image classification; deep-learning-based hyperspectral image classification methods [18] are based on two main directions. The first is basic spectral classification methods such as SVMs [19,20], k-nearest neighbor (KNN) [21,22], decision trees [23] and other classification algorithms based on spectral features and appropriate feature transformations. The second is to link image contextual features for classification. For example, spatial context and spectral correlation are utilized to enhance hyperspectral image classification [24], and a residual network [25] is employed to obtain features at each level and fuse deep features to the classification of the network [26]. In most dimensionality reduction classification methods, the data features retained by data dimensionality reduction are indistinguishable in classification, producing poor classification results. Determining the most suitable model for dimensionality reduction and classification of hyperspectral remote sensing data is crucial because the resolution of each pixel of hyperspectral remote sensing data is extremely high. Each pixel represents a large real area and mixes multiple terrains. If hard classification alone is too decisive, the model may fail to classify what proportion of terrain a single pixel occupies.

To solve these problems and improve the combination method of data dimensionality reduction and classification, this paper classifies hyperspectral urban terrain data that are subjected to various data dimensionality reduction algorithms. The supervised machine learning-based classification algorithm is used to observe which feature data after dimensionality reduction is sensitive to produce a strong classification effect. In addition, this study evaluates the overall accuracy of the combination of hyperspectral image dimensionality reduction and classification methods. Despite redundant data of hyperspectral remote sensing, the proportion of hyperspectral data features is preserved by different dimensionality reduction methods. The dimensionality reduction data retain the difference of the original features, and the supervised machine learning-based classification method is used to determine which data classification effect is better. The classification process is generally divided into hard and soft classification methods. This study also determines which dimensionality reduction and classification methods for hyperspectral remote sensing data are suitable for soft and hard classification.

The method determined in this study can quickly and accurately determine the terrain information of remote sensing images. Furthermore, the dimensionality of the overall data can be greatly reduced and redundant bands can be filtered, thereby increasing the calculation speed of remote sensing terrain analysis and retaining its maximum characteristics. Using lower-dimensional data to effectively express original high-dimensional features is conducive to the rapid extraction of information while compressing the amount of data. Soft classification and hard classification can be used for distinguishing pixel singularity for hyperspectral remote sensing terrain classification. In summary, this paper has two targets: (1) For the high-dimensional characteristics of hyperspectral data, various dimensionality reduction methods are applied to retain all the data features; then, it is verified whether unified manifold approximation and projection (UMAP) dimensionality reduction and SVM classification can both remove redundant data and maintain good classification accuracy. (2) For large-resolution hyperspectral images leading to hard pixel classification, a neural network model is applied to obtain the best fit of the pixel class decomposition.

## 2. Hyperspectral Dimensionality Reduction

### 2.1. Feature Extraction Dimensionality Reduction

Hyperspectral remote sensing images contain spectral information of all bands as well as considerable spatial and band redundancy. Hyperspectral feature extraction refers to the recombination and optimization of the original spectral–spatial features to extract new features that are most suitable for current application requirements. Hyperspectral feature extraction is based on the principle of reducing the data dimension to a lower level to achieve improved feature selection performance.

Based on the above, traditional machine learning methods can be applied for dimensionality reduction of hyperspectral image data because of the high number of spectra they contain and the high dimensionality of the data. Graph learning can effectively reveal the intrinsic relationships of data and is currently widely applied to hyperspectral imaging. For example, a hybrid graph learning method is applied to separate the intra-class compactness and inter-class separability of sample data and obtain the optimal spatial transformation for dimensionality reduction [27]. Graph learning considers only the individual information of each sample in certain features, and discriminant analysis-based hyperspectral image classification for dimensionality reduction [28] adds domain, tangential, and statistical features of samples to achieve feature complementarity and improve the classification performance of hyperspectral images. The difficulty of applying graph embedding methods to dimensionality reduction of hyperspectral images lies in choosing the appropriate neighborhood construction to explore spatial information. Spatial spectral manifold reconstruction preserving embedding [29] utilizes a new spatial and spectral combination distance (SSCD) to fuse spatial structure and spectral information to extract discriminative features. While deep learning convolutional neural networks can reduce the dimensionality of spectral and spatial data, the adoption of autoencoders in convolutional neural networks reduces the dimensionality [30], using pooled hyperspectral images [31].

This paper mainly focuses on the method of feature extraction based on machine learning for hyperspectral images. The existing feature extraction methods mainly include principal component analysis (PCA) [32], linear discriminant analysis (LDA) [33], factor analysis (FA) [34], singular value decomposition (SVD) [35], independent component analysis (ICA) [36], local linear embedding (LLE) [37], t-distributed stochastic neighbor embedding (t-SNE) [38], UMAP [39], and other dimensionality reduction methods. The data features retained by each method are different, and based on these differences, the feature extraction methods can be divided into linear dimensionality reduction and nonlinear dimensionality reduction. Nonlinear dimensionality reduction can be divided into global feature and local feature methods.

### 2.2. Linear Dimensionality Reduction

PCA is a linear transformation unsupervised dimensionality reduction algorithm. It maintains data information and simplifies the dimension of hyperspectral remote sensing data by transforming data information of all bands into a new coordinate system. In this method, the eigenvalue with the largest variance contribution is selected, which reduces the dimension of the dataset without much classification accuracy loss and achieves faster calculation. In view of the high-dimensional characteristics of hyperspectral data, the PCA algorithm selects multiple eigenvectors for feature extraction. The eigenvector selection rule based on experience is that when the number of principal components is greater than 1, the data with an eigenvalue greater than 1 and a variance ratio greater than 85% are selected.

LDA is a supervised linear dimensionality reduction algorithm; it projects high-dimensional pattern samples into the optimal discriminant vector space to extract classification information and compress the dimension of feature space. After projection, it is ensured that the pattern samples have the largest inter-class distance and the shortest intra-class distance in the new subspace; that is, the pattern has the best separability in this space. Classification aims to achieve the farthest possible distance between different categories and the shortest distance between the same category. Compared with the PCA

dimensionality reduction algorithm, LDA is more concerned with classification rather than maintaining data information. After dimensionality reduction, the intra-class variance of the data is the smallest, and the inter-class variance is the largest. The retained dimension is smaller than the classification category. When the number of layers of dimension reduction is three, the test dataset retains 99.9% of the original data features.

SVD dimensionality reduction is another feature extraction method similar to PCA. The method of considering the largest eigenvalue for Eigen decomposition is only suitable for the decomposition of square matrices but not for non-square matrices. Singular values, such as an m × n matrix, construct an m × m square matrix on the left, an n × n square matrix on the right, and an m × n singular matrix with non-zero diagonals in the middle; then, the values of the singular matrix are sorted by size for feature extraction. In the process of singular value decomposition, the singular value is reduced particularly fast, and fewer dimensions can be selected to retain more feature data.

ICA is another type of linear dimensionality reduction algorithm. It is suitable for the dimensionality reduction of non-Gaussian data because the signal obtained after the Gaussian distribution is mixed is Gaussian, and the mixed data show no difference. ICA is suitable for the separation of such mixed signals. For hyperspectral data, the mixed multiple spectra can be separated by ICA, extracting mutually independent attributes and thus reducing the dimension. The purpose of ICA is to extract mutually independent attributes and reduce dimensionality. It considers the data after dimensionality reduction to be a linear combination of several statistically independent components. It only focuses on independence and not the size of the data and the variance between them; therefore, it is more conducive to data separation.

### 2.3. Nonlinear Dimensionality Reduction

LLE is a nonlinear dimensionality reduction algorithm that preserves the local features of the data, allowing the dimensionality-reduced data to maintain a good manifold structure; hence, it is a manifold learning technology. LLE first measures the degree of linear correlation between each training instance and its nearest neighbors; then, the low-dimensional vector to represent the features of the training set is determined such that the local relationships are best preserved. When the LLE algorithm is used to reduce the dimensionality of the dataset to three-dimensional data, the dimensionality reduction error is $5.5 \times 10^{-18}$.

t-SNE is a nonlinear dimensionality reduction method. It usually pays more attention to maintaining similarity such that the distance between similar points in the low-dimensional space is smaller. The t-distribution has the characteristics of long tail; that is, when there are outliers, the entire distribution will not be separated from most of the original data because of the outliers. While t-distribution mapping is used for low-dimensional data, normal-distribution mapping is used for high-dimensional data. The t-SNE algorithm uses Gaussian distribution in a high-dimensional space and t-distribution in a low-dimensional space. The t-distribution satisfies the requirement for t-SNE that the distances between points in the same cluster are similar, whereas those between points in different clusters are large. t-SNE converts the similarity between data points into probabilities.

### 2.4. UMAP

UMAP is a nonlinear dimensionality reduction algorithm. It is based on the principle of manifold and projection technology to achieve dimensionality reduction. First, the distances between points in a high-dimensional space are calculated and projected to a low-dimensional space. Then, the distances between points in that low-dimensional space are calculated. Subsequently, stochastic gradient descent is used to minimize the difference between these distances. The most prominent feature of the UMAP output is the balance between local and global structure; UMAP tends to preserve the global structure in the final projection. It is used to analyze high-dimensional data of any data type, providing fast running time and high repeatability.

The UMAP dimensionality reduction algorithm mainly involves two steps—learning the manifold structure in the high-dimensional space and determining the low-dimensional representation method of the manifold.

Step 1: For a given original hyperspectral image dataset $X = (X_1, X_2, X_3 \cdots X_n)$, dimension reduction is initialized to obtain a low-dimensional dataset $Y = (Y_1, Y_2, Y_3 \cdots Y_n) \sim N(0, 10^{-4} \times I_n)$. Then, $p_{ij}$ and initial $q_{ij}$ are calculated.

The conditional probability of $i$ for $j$ is given by

$$p_{i|j} = e^{\frac{d(x_i, x_j)}{\sigma_i}} \tag{1}$$

The symmetric formula of the similarity matrix $P$ for $X$ is as follows:

$$p_{ij} = p_{i|j} + p_{j|i} - p_{i|j} \cdot p_{j|i} \tag{2}$$

The similarity matrix $Q$ of $Y$ is

$$q_{ij} = \left[ 1 + a \left( y_i - y_j \right)^{2b} \right]^{-1} \tag{3}$$

where $a \approx 1.93$ and $b \approx 0.79$ for default UMAP hyperparameters.

$p_{ij}$ measures the similarity between $X_i$ and $X_j$; $q_{ij}$ measures the similarity between $Y_i$ and $Y_j$.

Step 2: The cost loss is calculated using binary cross entropy (CE):

$$CE(p, q) = \sum_i \sum_j [p_{ij} \cdot log \frac{p_{ij}}{q_{ij}} + (1 - p_{ij}) \cdot log \left( \frac{1 - p_{ij}}{1 - q_{ij}} \right)] \tag{4}$$

Step 3: The parameters are optimized, and the number of iterations $t$, learning speed $v$, and momentum $a$ are set. The target result is a low-dimensional data representation $Y = Y_1, Y_2, Y_3 \cdots Y_n$.

Step 4: Optimization is started: $Y^t = Y^{t-1} + v \cdot \frac{dC}{dY} + a \cdot \left( Y^{t-1} - Y^{t-2} \right)$, where $\frac{dC}{dY}$ is the gradient vector of the loss function with respect to $Y$; $\frac{dC}{dY} = \left( \frac{\partial C}{\partial y_i} \right)_{1 \times n}$.

In the field of remote sensing, PCA is commonly used for feature dimensionality reduction and machine learning for classification. Considering the diversity of different regions of a superpixel-level PCA for dimensionality reduction [40], a novel collapsed PCA was developed [41], in which the spectral vector is collapsed into a matrix to effectively determine the covariance matrix for better dimensionality reduction of hyperspectral image data. The results of different types of PCA and their linear variants were compared, and it was found that the segmented PCA has better accuracy than the collapsed PCA, which has a lower spatial complexity [42]. Because PCA projection looks for the direction that maximizes the variance, it usually ignores the local structure and the variance in other directions. UMAP in machine learning, on the other hand, differs from PCA in that not only local features but also global features are preserved when clustering.

Remote sensing data contains hundreds of bands, and the combination of each band constitutes the pixel point category of image data. The input image is categorized according to the type of each pixel point, and the UMAP algorithm is applied to reduce the dimensionality of hyperspectral linear data. UMAP maps the original band data to the low-dimensional space, for which it first learns the popular structure of hyperspectral nonlinearity to determine the low-dimensional representation of the stream shape. The optimal low-dimensional data are found by utilizing the least cost function, which results in dimensionality reduction. Unlike other dimensionality reduction algorithms, such as PCA and t-SNE, which only consider the local structure and lose many features, UMAP considers the global structure.

## 3. Hyperspectral Image Classification Methods

The current difficulties to be overcome by feature extraction methods in the field of remote sensing image research are the large spatial feature span of hyperspectral images and the limited number of samples. CNNs are widely applied for hyperspectral image classification [43], and the insufficient number of samples of hyperspectral images is a bottleneck that limits the deep learning training, for example, utilizing knowledge migration [44] and domain generalization [45] approaches to address the characteristics of the insufficient number of samples. Hyperspectral image classification using deep neural networks with inadequate samples and feature learning [46]. A new method was proposed for hyperspectral image classification based on multi-view deep neural networks, which fuses spectral and spatial features, employing only a small number of labeled samples [47]. Multi-level discontinuous features are extracted from remote sensing images [48]. Moreover, hyperspectral images are classified using a regularized subspace of Manhattan distances [49] by introducing a deep hybrid multi-view neural network that implements information interaction, making excellent utilization of different graph filters [50].

In the face of hyperspectral image classification, both hard classification and soft classification are applied. Hard classification mainly involves deep dimensionality reduction of hyperspectral images first, followed by supervised classification by machine learning. Soft classification is the probability obtained from image classification using logistic regression and neural networks and represents the decomposition of hyperspectral image elements.

### 3.1. Hard and Soft Classification for Hyperspectral Images

Hard classification and soft classification are common strategies for hyperspectral remote sensing image classification. Figures 1 and 2 show the flowcharts of hard and soft classification, respectively. Hard classification assigns each pixel in a remote sensing image a single category, and the classification is based on the similarity of pixel features, spectral features, texture features, or a mixture of multiple features with known statistical features of each category.

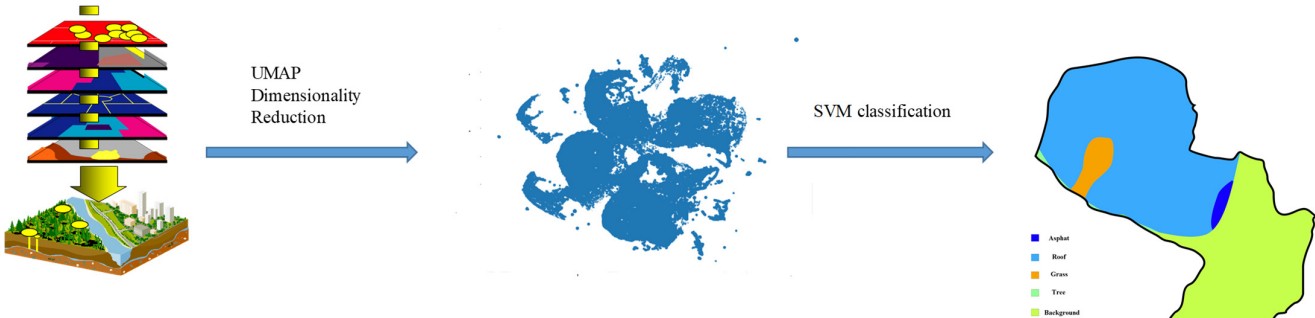

**Figure 1.** Flowchart of hard classification.

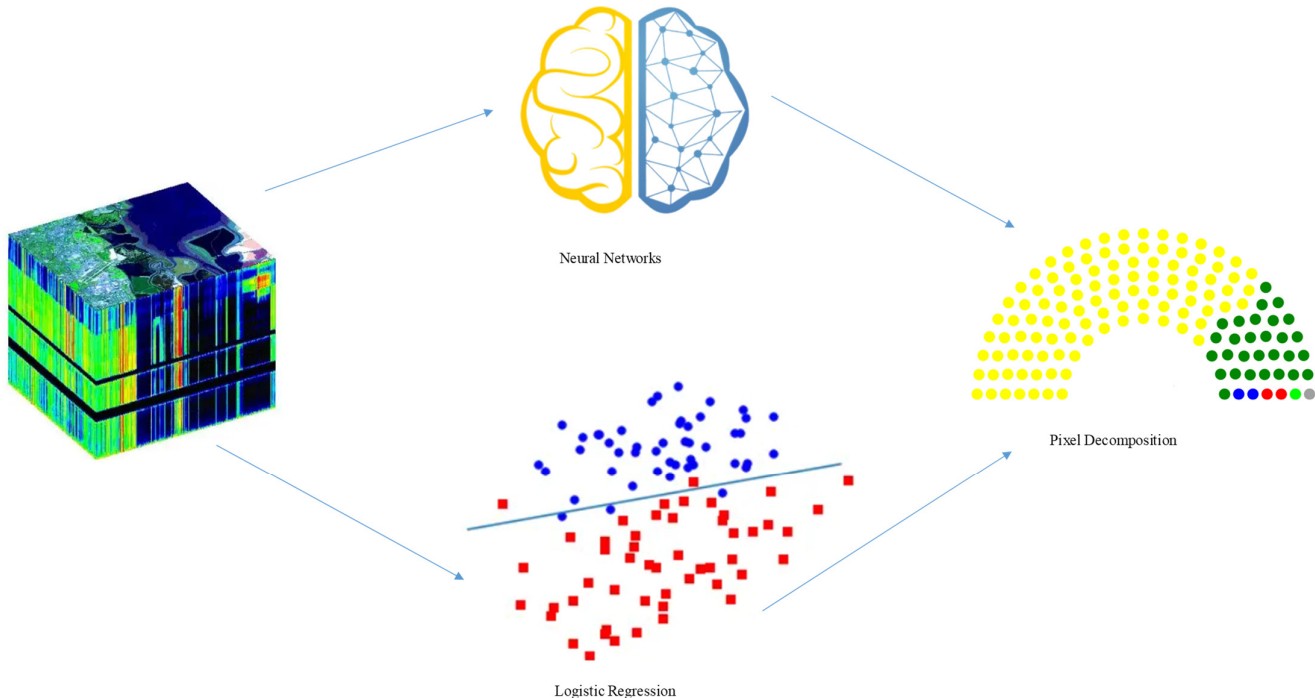

**Figure 2.** Flowchart of soft classification.

In soft classification, the surface area corresponding to the pixel is often composed of multiple categories of ground objects according to the actual situation, therefore it is assumed that each pixel belongs to multiple categories or is composed of multiple categories. Then, the relationship between the pixel and each category is calculated according to a specific algorithm. The classification output is the probability fuzzy classification that the pixel belongs to each category or the proportion of each category of objects in the pixel.

Soft classification and hard classification have their own advantages in the classification of hyperspectral remote sensing images. Hard classification focuses only on the maximum value of the corresponding classification probability of each pixel, ignoring the relationship between other surrounding pixels and the corresponding relationship. Soft classification can divide a single pixel into the proportion of target objects. The hard classification method is absolute; therefore, it determines the category of an object according to the category of its adjacent objects.

The dimensionality reduction operation is performed on the basis of the distribution characteristics of hyperspectral data. In this study, the data are obtained from public datasets with data labels. The terrain classification methods of hyperspectral images mainly include supervised machine learning, KNN, Bayesian classification, logistic regression, neural network, SVM, and decision tree classification methods. Bayesian and logistic regression usually have better classification effect on linear features. The KNN algorithm determines the object category according to the distance of adjacent nearest points and provides a high classification speed and good prediction effect. The decision tree algorithm can process multiple continuous fields and can effectively classify according to the intensity of hyperspectral bands. The SVM can solve the nonlinear problem of hyperspectral light intensity, providing relatively strong generalization ability and handling high-dimensional data effectively. The neural network can be fitted by multi-dimensional parameters and has strong learning ability and high classification accuracy.

Soft classification of hyperspectral images is a probabilistic model in which the reduced-dimensional data features are classified by a logistic regression or neural network model, yielding a probability that a single pixel on the image can be simultaneously classified into multiple categories; this yields a probability that the pixel is classified into each category, which can be decomposed by the probability for image elements. The hard

classification of hyperspectral images is a non-probabilistic model, i.e., the results are derived by a decision classification function, and the exact classification of each pixel point is learned by a supervised classifier of machine learning for image distribution features.

### 3.2. Neural Networks

Feedforward neural networks are also known as multi-layer perceptrons, in which different neurons belong to different layers, namely an input layer, a hidden layer, and an output layer, which are fully connected to each other. Figure 3 illustrates a neural network structure. The neural network contains functions such as activation and loss functions. For the classification of hyperspectral images, the activation function is selected as tanh, the Adam gradient descent function is used, the number of nodes in the hidden layer is selected as 50, and the number of iterations is selected as 5000 for training. The neuron is directly connected to the following formula:

$$y_i = \sum_j w_{ij} x_j + b \tag{5}$$

where $W$ is the weight and $b$ is bias.

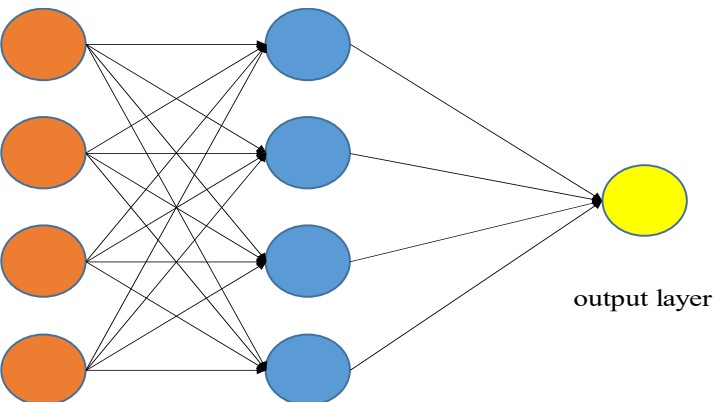

**Figure 3.** Neural network structure diagram.

The tanh activation function is obtained as

$$\tanh(x) = \frac{e^x - e^{-x}}{e^x + e^{-x}} \tag{6}$$

### 3.3. Support Vector Machine

SVM is one of the classification algorithms in supervised learning, whose core idea is to determine a divided hyperplane that separates sample points in space; the plane with the largest distance to the nearest point among the samples of different categories with the best generalization ability is selected as the divided hyperplane.

The point-to-plane distance $\gamma$ is obtained as

$$\gamma = \frac{|w^T x_i + b|}{||w||} \tag{7}$$

where $w$ is the weight and $b$ is the bias.

The point with the smallest sample interval is found to have the largest distance to the plane.

The nonlinear data of hyperspectral images are not in the same low-dimensional space and need to be mapped to the high-dimensional space by applying kernel functions so that linearly indistinguishable sample points can be made linearly distinguishable.

## 4. Experimental Results and Analysis

### 4.1. Hyperspectral Image Description

The dataset used in this study is a public dataset comprising hyperspectral images of a city obtained by the HYDICE sensor. The image size is 307 × 307 pixels. The urban dataset originally has a total of 210 bands; after removing noise and water absorption bands, 162 bands remain for subsequent dimensionality reduction processing and analysis. There are four types of ground objects: asphalt, roof, grass, and tree. The training set and test set are in the ratio of 8:2. The test set is (75,399, 18,850). The number of pixels in each test set and training set is shown in Table 1. An original hyperspectral remote sensing image and its label image showing different terrains are shown in Figure 4.

**Table 1.** Train and test sets of the dataset.

|           | Asphalt | Grass  | Tree   | Roof |
|-----------|---------|--------|--------|------|
| Number    | 29,954  | 32,328 | 24,805 | 7162 |
| Train set | 23,963  | 25,910 | 19,766 | 5760 |
| Test set  | 5991    | 6418   | 5039   | 1402 |

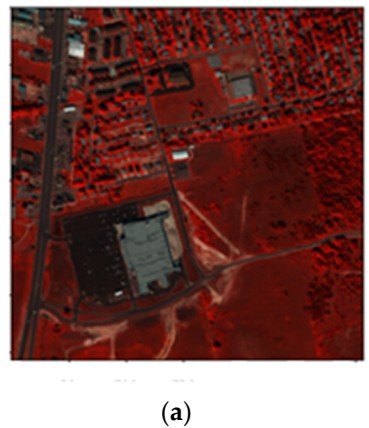
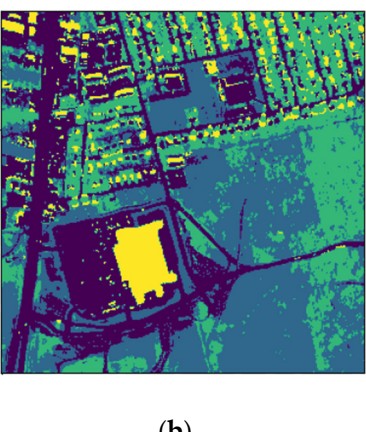

(**a**)　　　　　　　　　　　　　　　　　　　　(**b**)

**Figure 4.** (**a**) Hyperspectral remote sensing image; (**b**) label image showing different terrains.

Hyperspectral images have high resolution and rich spectral information; however, the band interval between each spectrum is small, a nonlinear relationship exists between each spectrum, and the large amount of data results in abundant redundant data, making identification difficult. Through dimensionality reduction of hyperspectral remote sensing images, data that retains the overall feature correlation according to the distribution characteristics of the data can be extracted, thereby drastically reducing the data dimension.

### 4.2. Results and Analysis

#### 4.2.1. PCA

PCA is applied to the urban hyperspectral image data, with 162 bands and features, to reduce the dimension, and the ratio of the features is sorted from the largest to the smallest. The data dimension retained by PCA dimensionality-reduced data and the variance ratio of the occupied features are shown in Figure 5.

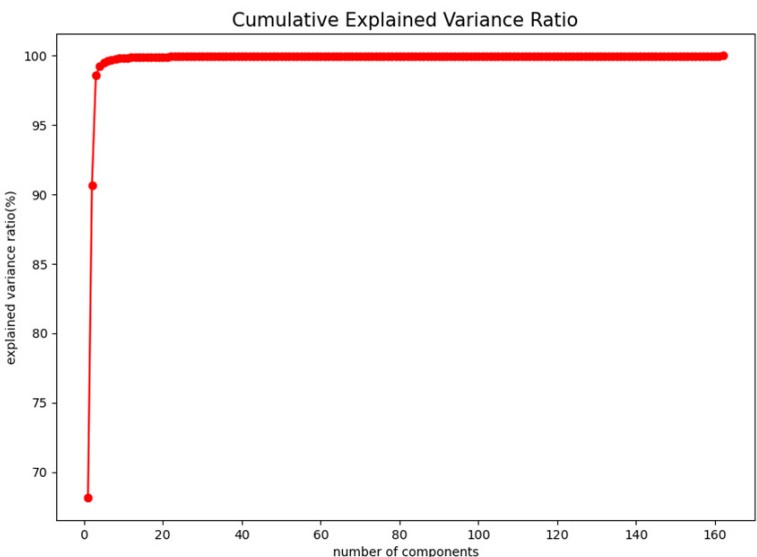

**Figure 5.** PCA cumulative explained variance ratio.

The figure indicates that the first and second vectors account for more than 60% and 30% of the total proportion. A PCA dimensionality reduction dimension of the largest three features accounts for 98% of the total variance ratio. For each additional dimension, the proportion of features occupied by data increases slowly. When the classification dimension is four, the proportion of features reaches 99%. Thus, the purpose of reducing the amount of data and preserving the overall characteristics is met.

### 4.2.2. KNN Proximity Classification

The choice of the k value of the KNN will have a considerable effect on the algorithm classification. When k = 1, the test instance is related to the closest sample, and the training error is small. By contrast, when the test sample has noise, the test error will be large. While a small k value results in overfitting, when the k value is large, it is equivalent to training with a large range of data, and the test result is the class with the most instances in the range, which will cause underfitting. Thus, selecting an appropriate k value is important. As the test sample, we consider the overall data of the test set without dimensionality reduction. The k value and the segmentation accuracy are selected as shown in Table 2.

**Table 2.** Selection of KNN classification k.

| KNN | k = 3 | k = 4 | k = 5 |
|---|---|---|---|
| AA | 0.9587 | 0.9754 | 0.9751 |
| OA | 0.9564 | 0.9770 | 0.9762 |
| KAPPA | 0.9382 | 0.9675 | 0.9664 |
| RECALL | 0.9600 | 0.9755 | 0.9752 |
| F1-SCORE | 0.9593 | 0.9754 | 0.9751 |

When k = 4, the feature ratio reaches 99%, and the overall classification accuracy is somewhat improved compared with that when k = 3. When the k value increases, the calculation amount increases and the classification accuracy decreases. In the KNN proximity algorithm, four nearby points are selected as the criteria for their classification.

### 4.2.3. Gaussian Maximum Likelihood Classifier

The KNN proximity algorithm only considers the distance from the sample to be classified to the center of each sample, ignoring the overall distribution of the sample. By

contrast, the Gaussian maximum likelihood classifier additionally considers the distribution characteristics of known classes as well. The classifier is run on the hyperspectral image dataset to train the reduced principal components and classify them according to the training data. The classified images are shown in Figure 6. The classification accuracy of each terrain is shown in Table 3.

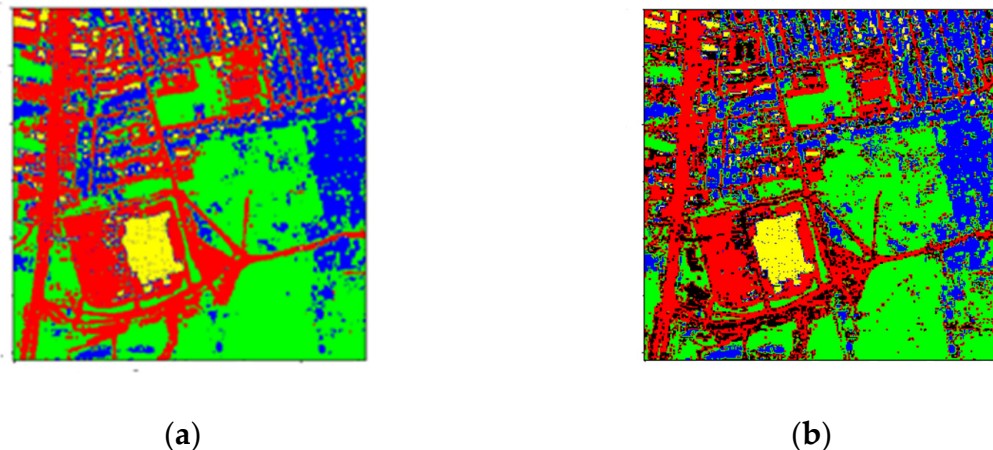

(**a**)  (**b**)

**Figure 6.** (**a**) Classified image of Gaussian maximum likelihood; (**b**) Result of comparison with the label image.

**Table 3.** Accuracy of Gaussian Maximum Likelihood Classification.

|  | Asphalt | Grass | Tree | Roof |
|---|---|---|---|---|
| PA | 0.8502 | 0.9029 | 0.8968 | 0.9139 |
| OA | 0.9234 | 0.8814 | 0.9137 | 0.7108 |
| F1-Score | 0.8853 | 0.8920 | 0.9051 | 0.7996 |

As the figure shows, in the classification result of the Gaussian maximum likelihood classifier after dimensionality reduction, the black pixels are points that are classified incorrectly. Overall, a reasonable classification effect is achieved. The table indicates that the best classification accuracy of Gaussian maximum likelihood estimation for each terrain is 91.39%. However, the accuracy required for terrain segmentation of hyperspectral remote sensing data is insufficient, and the Gaussian maximum likelihood classifier cannot distinguish the spectral features retained by PCA dimensionality reduction well.

4.2.4. Dimensionality Reduction Method Combined with Classification

Next, we determine an effective classification algorithm to identify dimensionality-reduced data with multiple features retained by different dimensionality reduction methods. After dimensionality reduction, the data features include linear features, local features, manifold features, and non-popular features. In addition, the classifiers are divided into linear and nonlinear classifiers. A suitable method for dimensionality reduction and classification of hyperspectral images can be determined through experiments. The classification results of various dimensionality reduction and classification methods are shown in Table 4, and the classification accuracy is shown in Figure 7.

**Table 4.** Classification accuracy of hyperspectral images with different dimensionality reduction classification methods.

| | | None Dimensionality Reduction | PCA | LDA | LLE | T-SNE | SVD | ICA | FA | UMAP |
|---|---|---|---|---|---|---|---|---|---|---|
| k-Nearest Neighbor | Kappa | 0.9612 | 0.9674 | 0.9126 | 0.9460 | 0.9322 | 0.9659 | 0.9401 | 0.9479 | 0.9938 |
| | Recall | 0.9690 | 0.9755 | 0.9314 | 0.9642 | 0.9468 | 0.9745 | 0.9617 | 0.9608 | 0.9957 |
| | AA | 0.9736 | 0.9754 | 0.9297 | 0.9631 | 0.9499 | 0.9750 | 0.9604 | 0.9593 | 0.9987 |
| | F1-score | 0.9712 | 0.9754 | 0.9306 | 0.9636 | 0.9483 | 0.9747 | 0.9610 | 0.9600 | 0.9957 |
| | OA | 0.9729 | 0.9770 | 0.9382 | 0.9618 | 0.9521 | 0.9759 | 0.9577 | 0.9631 | 0.9956 |
| | P | | 0.1445 | $2 \times 10^{-5}$ | 0.044 | 0.0005 | 0.2471 | 0.0205 | 0.0114 | $4 \times 10^{-6}$ |
| Naive Bayesian Classifier | Kappa | 0.5269 | 0.8200 | 0.8181 | 0.5237 | 0.4931 | 0.8190 | 0.4751 | 0.7392 | 0.2816 |
| | Recall | 0.6128 | 0.8702 | 0.8859 | 0.6346 | 0.5271 | 0.8692 | 0.6346 | 0.7843 | 0.3961 |
| | AA | 0.6358 | 0.8441 | 0.8253 | 0.6923 | 0.4901 | 0.8436 | 0.6661 | 0.7902 | 0.3972 |
| | F1-score | 0.6070 | 0.8551 | 0.8410 | 0.6405 | 0.5035 | 0.8544 | 0.5998 | 0.7833 | 0.3369 |
| | OA | 0.6617 | 0.8721 | 0.8692 | 0.6708 | 0.6494 | 0.9714 | 0.6285 | 0.8165 | 0.5167 |
| | P | | $9 \times 10^{-6}$ | $1 \times 10^{-6}$ | 0.5411 | 0.0767 | $6 \times 10^{-5}$ | 0.8466 | 0.0001 | 0.0011 |
| Support Vector Machine | Kappa | 0.9790 | 0.9796 | 0.9228 | 0.4850 | 0.9127 | 0.9819 | 0.7803 | 0.9763 | 0.9937 |
| | Recall | 0.9852 | 0.9855 | 0.9371 | 0.5092 | 0.9312 | 0.9862 | 0.7492 | 0.9839 | 0.9958 |
| | AA | 0.9845 | 0.9832 | 0.9375 | 0.7552 | 0.9328 | 0.9855 | 0.8908 | 0.9822 | 0.9957 |
| | F1-score | 0.9848 | 0.9843 | 0.9373 | 0.4542 | 0.9319 | 0.9859 | 0.7795 | 0.9831 | 0.9957 |
| | OA | 0.9851 | 0.9856 | 0.9455 | 0.6544 | 0.9384 | 0.9872 | 0.8485 | 0.9833 | 0.9955 |
| | P | | 0.8780 | $1 \times 10^{-6}$ | 0.0001 | $2 \times 10^{-6}$ | 0.2953 | 0.0002 | 0.3226 | $1 \times 10^{-5}$ |
| Decision Tree | Kappa | 0.9374 | 0.9562 | 0.8855 | 0.9300 | 0.9116 | 0.9483 | 0.9208 | 0.9488 | 0.9890 |
| | Recall | 0.9540 | 0.9682 | 0.9075 | 0.9546 | 0.9346 | 0.9627 | 0.9477 | 0.9316 | 0.9923 |
| | AA | 0.9520 | 0.9656 | 0.9049 | 0.9504 | 0.9331 | 0.9596 | 0.9464 | 0.9589 | 0.9927 |
| | F1-score | 0.9529 | 0.9669 | 0.9062 | 0.9525 | 0.9338 | 0.9612 | 0.9469 | 0.9603 | 0.9925 |
| | OA | 0.9558 | 0.9691 | 0.9191 | 0.9506 | 0.9376 | 0.9635 | 0.9441 | 0.9639 | 0.9922 |
| | P | | 0.0065 | $9 \times 10^{-5}$ | 0.6285 | 0.0078 | 0.0806 | 0.1688 | 0.7430 | $2 \times 10^{-6}$ |
| Logistic Regression | Kappa | 0.9491 | 0.9451 | 0.9167 | 0.4480 | 0.4663 | 0.9459 | 0.5427 | 0.9468 | 0.5121 |
| | Recall | 0.9597 | 0.9569 | 0.9326 | 0.4804 | 0.5416 | 0.9569 | 0.5372 | 0.9571 | 0.5308 |
| | AA | 0.9588 | 0.9576 | 0.9325 | 0.6262 | 0.5754 | 0.9574 | 0.5875 | 0.9582 | 0.4935 |
| | F1-score | 0.9592 | 0.9572 | 0.9326 | 0.3946 | 0.5483 | 0.9572 | 0.4979 | 0.9576 | 0.5106 |
| | OA | 0.9641 | 0.9613 | 0.9412 | 0.6308 | 0.6283 | 0.9618 | 0.6923 | 0.9624 | 0.6650 |
| | P | | 0.5074 | 0.0004 | $2 \times 10^{-5}$ | $3 \times 10^{-7}$ | 0.5353 | $3 \times 10^{-6}$ | 0.6350 | $1 \times 10^{-8}$ |
| Multi-layer Perceptron | Kappa | 0.9701 | 0.9666 | 0.9197 | 0.8274 | 0.8924 | 0.9784 | 0.9154 | 0.9749 | 0.9748 |
| | Recall | 0.9814 | 0.9772 | 0.9362 | 0.8042 | 0.9142 | 0.9854 | 0.9281 | 0.9832 | 0.9786 |
| | AA | 0.9751 | 0.9743 | 0.9330 | 0.9002 | 0.9194 | 0.9835 | 0.9441 | 0.9819 | 0.9809 |
| | F1-score | 0.9782 | 0.9757 | 0.9346 | 0.8322 | 0.9168 | 0.9844 | 0.9356 | 0.9826 | 0.9797 |
| | OA | 0.9789 | 0.9764 | 0.9432 | 0.8798 | 0.9241 | 0.9847 | 0.9404 | 0.9823 | 0.9822 |
| | P | | 0.3516 | $8 \times 10^{-6}$ | $9 \times 10^{-5}$ | $5 \times 10^{-6}$ | 0.0222 | $4 \times 10^{-5}$ | 0.1249 | 0.3116 |

We analyzed the adaptation of different classification algorithms to the features retained by the dimensionality reduction methods. For the KNN proximity algorithm, the data features retained by UMAP dimensionality reduction can be well captured. Comparison of the classification accuracy of FA and SVD dimensionality reduction with the classification accuracy of logistic regression indicates that they are comparable to unreduced dimensionality; therefore, it can be concluded that the dimensionality reduction not only reduces the amount of data but also achieves good classification accuracy. The Naive Bayes

algorithm and SVD have the best classification accuracy of hyperspectral data compared with other dimensionality reduction methods. The classification accuracy of SVM and UMAP dimensionality reduction is the best among all dimensionality reduction algorithms. The multi-layer perceptron algorithm is not sensitive to LLE dimensionality reduction. The combination of multi-spectral classification with SVD dimensionality reduction has the best effect compared with other combination methods. The combination method of decision tree classification and UMAP dimensionality reduction shows superior results. The accuracy of classification with the dimensionality reduction algorithm and that without dimensionality reduction were statistically analyzed, and the *p*-value was less than 0.05, indicating a significant difference.

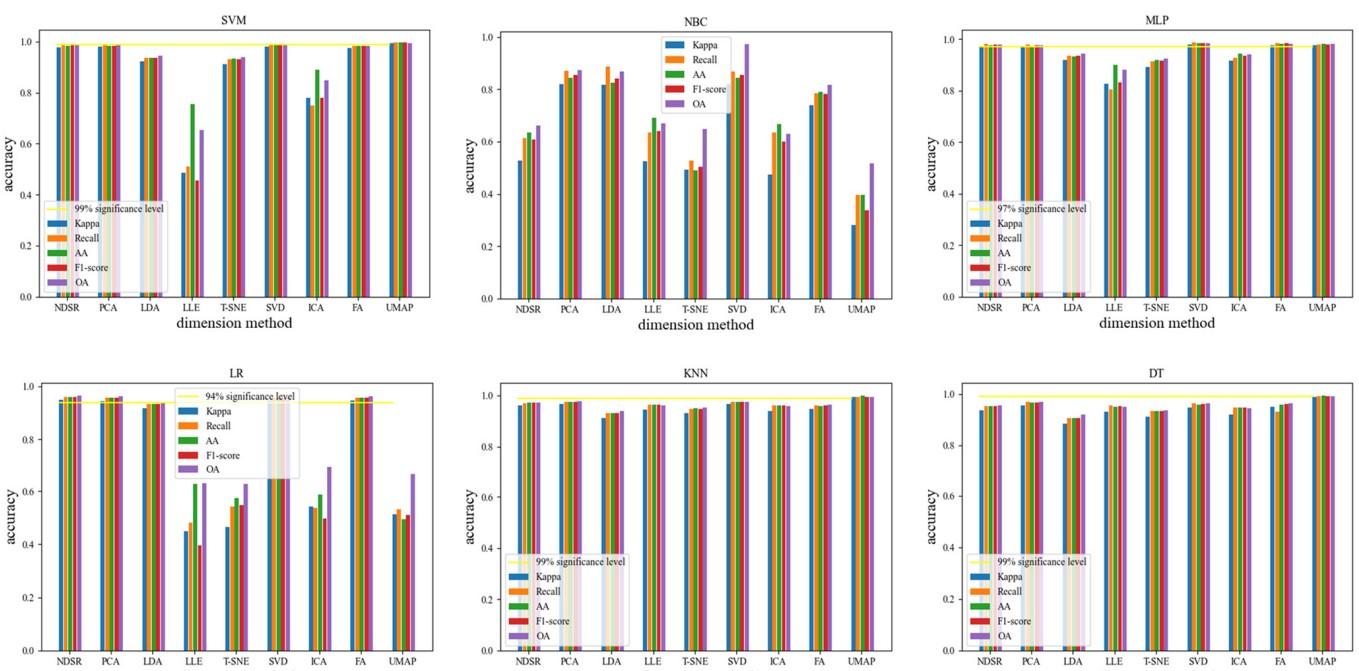

**Figure 7.** Comparison chart of overall classification accuracy of different dimensionality reduction methods.

### 4.2.5. Accuracy of Various Dimensionality Reduction Methods for Each Terrain Classification

The accuracy of each terrain classification obtained by the dimensionality reduction method and the classification method is shown in Figure 8.

Clearly, the Naive Bayes data classification effect on dimensionality reduction is not as good as that of other classification algorithms. Compared with the classification accuracy of algorithms without dimensionality reduction, the dimensionality reduction algorithms achieve the expected goal of not only reducing the amount of calculation but also preserving the original classification effect.

The dimensionality reduction mode of UMAP preserves the integrity of the data features. In the case of the Naive Bayes algorithm, the classification accuracy based on t-SNE dimensionality reduction for roof terrain is 0, because the remote sensing spectra are nonlinear, independent, and discrete. According to the results, the Bayesian classifier is not highly effective for handling the nonlinear problem of hyperspectral data. The comparison chart of terrain classification accuracy by UMAP dimensionality reduction is shown Figure 9.

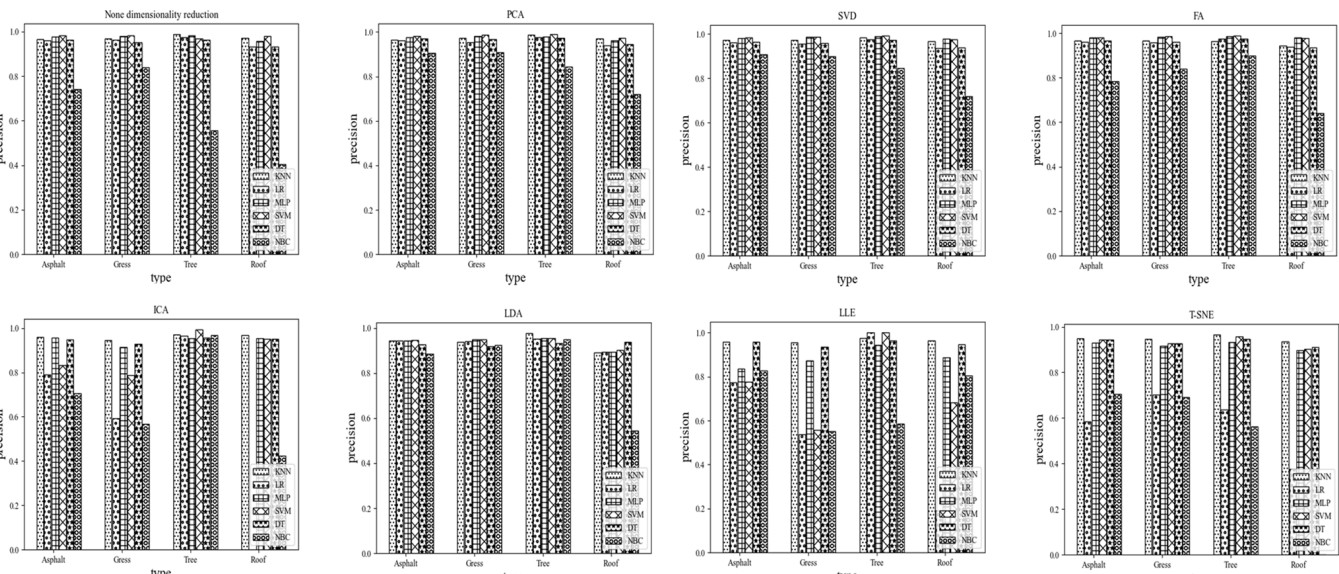

**Figure 8.** Comparison chart of terrain classification accuracy by different dimensionality reduction methods.

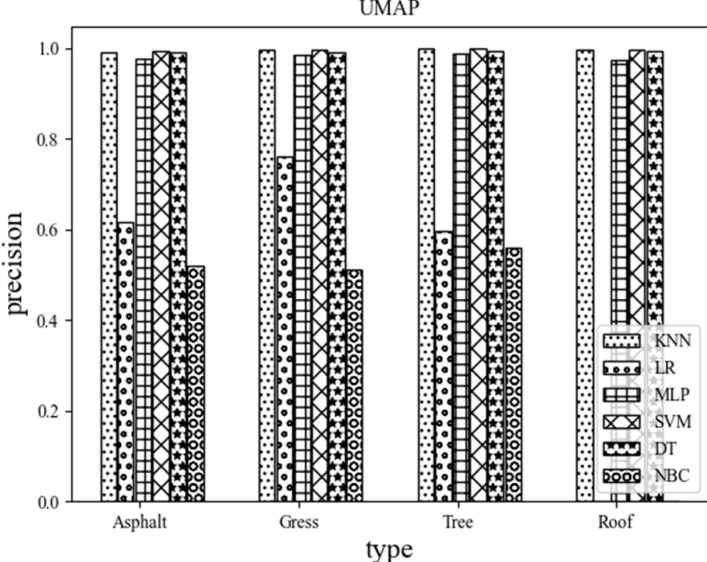

**Figure 9.** Comparison chart of terrain classification accuracy by UMAP dimensionality reduction.

The nonlinear local features retained by UMAP are not sensitive to the classification effect of linear logistic regression for some linear classifiers such as the Bayesian classifier. However, the classification accuracy obtained by UMAP dimensionality reduction with other classifiers can reach more than 99%. UMAP has the ability to infer local and global structures while maintaining the relative global distance in the low-dimensional space. UMAP not only has high classification accuracy but also reduces the running time. With its high accuracy, UMAP provides favorable technical support for the classification of hyperspectral remote sensing terrains.

### 4.3. Soft Classification of Hyperspectral Images

The classification of hyperspectral remote sensing images is restricted by its spatial resolution. The spatial resolution refers to the size of the smallest target object that the sensor can distinguish; it is the ground range corresponding to one pixel in the actual satellite observation image. Each pixel of the image represents the real area of land, and

the image will face various terrains on its pixels. Most of the classification prioritizes the type of land, ignoring the texture and structure of the remote sensing image feature. The topography of pixel elements is decomposed into probabilities for soft classification of hyperspectral images.

The following two methods are used to fit the correct classification model. First, the soft classification model of pixels is trained according to the overall spectral data, and then the multiple logistic regression and neural network models will be fitted. One pixel of the image contains the range of the square of the real object, and the side length of the square depends on the image resolution at the time of capturing the image. For a pixel mixed with multiple terrains, the proportion of each type of terrain in a single pixel is obtained. The topographic ratio of the pixel classification determines the local image classification.

Using a linear logistic regression function with hyperspectral remote sensing data as input, the targets are classified into terrain types. Multi-variate logistic regression is performed, and the output is the probability ratio of each class. Logistic regression and neural network training are performed for 162 bands of remote sensing data as input parameters. In the original dataset, each pixel is divided into proportions of target objects, and the error calculation and fitting are performed between the probability result of logistic regression and the true proportion. The fitting results and errors obtained are shown in Table 5.

**Table 5.** Soft classification fitting accuracy.

| | RMSE | $R^2$ | $p$-Value |
|---|---|---|---|
| Linear Logistic Regression | 0.103 | 0.709 | 0.869 |
| Neural Networks | 0.042 | 0.979 | 0.071 |

Table 5 indicates that the fitting degree of the neural network is smaller than that of the linear logistic regression. The multi-layer neurons of the neural network can fit the nonlinear data characteristics between the spectra well, and the pixel classification of the hyperspectral image can obtain the proportion of terrain categories through the soft classification probability of the output of the neural network. The spatial characteristics of hyperspectral images can be enhanced according to the adjacent pixel types, thus providing support for efficient classification accuracy improvement in hyperspectral images.

The statistical significance test comparing the classification results of different methods also suggests that linear regression downscaling is not relevant for hyperspectral downscaling analysis, and the linear regression fit is insufficient because of the nonlinearity of the spectra. However, the neural network has the best fit and correlation effect.

## 5. Discussion

The UMAP dimensionality reduction and SVM classification methods for hyperspectral images can retain and extract the main features. Statistical significance tests are performed for the results. The computational effort of UMAP dimensionality reduction and SVM classification of hyperspectral images is low and their accuracy of classification is improved. Considering the high pixel resolution of hyperspectral images and the complexity of image elements, the pixel decomposition of the image and the fitting of the neural network model can solve the problem of excessive classification errors of image pixels due to a high image resolution.

The nearest regularized subspace (NRS) classification method is applied to determine the residuals between the approximation and the corresponding pixels, and the Tikhonov matrix of each class and the pixels to be classified into the corresponding classes. In determining the best distance metric [51], this method achieves a maximum accuracy of 96% but fails to reduce the number of parameters computed for hyperspectral images. When the combined hyperspectral image and LiDAR data are used, LiDAR data help to better characterize the elevation information of the same measurement area, thereby

improving the classification performance [52]. The fusion of CNN networks and LiDAR data to fuse feature- and decision-level classification achieved good accuracy in the test set [53]. Thus, this fusion strategy can be applied for pixel classification of hyperspectral images. Hyperspectral dimensionality reduction classification using QPCA [54] and deep learning networks combined with meta-learning, as well as population intelligence and evolutionary algorithms (SIEA), can be used to solve the feature selection problem for hyperspectral images [55]. Although all the above-mentioned methods preserve the main features of hyperspectral data, they are inferior to UMAP, which preserves the global features of the images.

Here, we selected supervised classification methods that are highly dependent on the quality of the sample labels of the dataset and require a small sample size and training feature classification for fewer samples. All the dimensionality reduction methods involve feature transformation and extraction of data by ignoring the connection between spectra. Deep learning networks can help both dimensionality reduction and image classification of hyperspectral images. In our future work, we will consider combining deep learning for hyperspectral information classification.

## 6. Conclusions

In this study, the combination of feature extraction dimension reduction and classification of hyperspectral remote sensing images was investigated. The analysis results indicate that the UMAP data dimension reduction algorithm and the SVM classification algorithm achieve superior performance in the terrain classification of hyperspectral images. A classification accuracy of 99.57% could be achieved. The SVM algorithm effectively captures the classification features after dimensionality reduction, and it not only reduces the amount of calculation but also improves the classification accuracy. For the single-pixel classification problem of hyperspectral images, it is advisable to not rely solely on the maximum probability. The neural network fitting classification model was found to be the most effective for fitting the classification probability of pixels according to the spectral features; it achieved a fitting correlation coefficient ($R^2$) of 0.979. It can provide a method for solving the single-pixel classification problem. To prepare for the classification and recognition of hyperspectral data and ground objects, the topographic and edge features of ground objects can be identified on the basis of the results of soft classification, which improves identification and classification accuracy. This study emphasizes that deep learning neural networks for image classification are expected to be widely applied to hyperspectral images.

**Author Contributions:** Conceptualization, H.L. and L.C.; methodology, L.C. and X.Z.; software, H.L. and J.C.; validation, L.C., Y.H. and H.L.; writing—original draft preparation, H.L.; writing—review and editing, L.C. and JC. All authors have read and agreed to the published version of the manuscript.

**Funding:** This paper was supported by the National Natural Science Foundation of China (Grant No. U19A2061), the 2021 Jilin Provincial Budget Construction Fund (Innovation Capacity Development) (Grant No. 2021C044-4), and the 2021 Science and Technology Research Project of Jilin Provincial Department of Education (Grant No. JJKH20210337KJ).

**Data Availability Statement:** Not applicable.

**Conflicts of Interest:** The authors declare no competing interests.

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
