# Peer review of "Dimensionality Reduction and Classification of Hyperspectral Remote Sensing Image Feature Extraction"

_remotesensing, doi:10.3390/rs14184579_

Round 1

Reviewer 1 Report (Previous Reviewer 1)

Major comments:

1.       The title is confusing. Please revise for clarity.

2.       The novelty of the proposed paper and its contribution to the research field  must be explicitly stated.

3.       Discussion of hyperspectral image dimensionality reduction and classification methods focuses on traditional (classical) techniques. More recent deep learning based techniques and the recently introduced nearest regularized subspace (NRS) classifier also should be discussed. The authors are encouraged to discuss, among other papers, Khan, et al. (2022). Hyperspectral image classification using NRS with different distance measurement techniques. Multimedia Tools and Applications, 81(17); Roy, et al. (2022). Hyperspectral and LiDAR data classification using joint CNNs and morphological feature learning. IEEE Transactions on Geoscience and Remote Sensing, 60.

4.       Add the workflow diagram to explain your methodology and its steps.

5.       Add the discussion section to discuss the limitations of the proposed methodology.

Minor (technical comments):

1.       Remove axis labels and values from photo images presented in Figures 2 and 4.

2.       All equations must have their variables explained. Currently, it is missing, for example, in Equation 7.

3.       Figure 6: labels and text font size is too small. The figure is unreadable.

Author Response

Point 1: The title is confusing. Please revise for clarity.

Response 1: Thank you for your comment.We changed the title to Dimensionality Reduction and Classification of Hyperspectral Remote Sensing Image Feature Extraction. Recent methods for downscaling and classification of hyperspectral images are presented.

Point 2: The novelty of the proposed paper and its contribution to the research field must be explicitly stated.

Response 2: Thank you for your comment. We summarize the results of the experiment and mark them in the third paragraph of the introduction.

Point 3: Discussion of hyperspectral image dimensionality reduction and classification methods focuses on traditional (classical) techniques. More recent deep learning based techniques and the recently introduced nearest regularized subspace (NRS) classifier also should be discussed. The authors are encouraged to discuss, among other papers, Khan, et al. (2022). Hyperspectral image classification using NRS with different distance measurement techniques. Multimedia Tools and Applications, 81(17); Roy, et al. (2022). Hyperspectral and LiDAR data classification using joint CNNs and morphological feature learning. IEEE Transactions on Geoscience and Remote Sensing, 60.

Response 3: Thank you for your comment. We discuss the latest articles related to hyperspectral classification, the application of NRS in classification and the feature fusion strategy of CNN and LiDAR. We mark the modified parts as yellow in the second paragraph of the discussion section.

Point 4: Add the workflow diagram to explain your methodology and its steps.

Response 4: Thank you for your comment. The flow chart of soft and hard classification is shown in Figure 1 and Figure 2.

Point 5: Add the discussion section to discuss the limitations of the proposed methodology.

Minor (technical comments):

Response 5: Thank you for your comment.We have added a discussion section describing the experimental results of the method in this paper, a comparison with the latest techniques, and also a discussion of the limitations of the current method.

Point 6: Remove axis labels and values from photo images presented in Figures 2 and 4.

Response 6: Thank you for your comment.We have removed the labels and values from the photo images in Figure 4 and Figure 6.

Point 7: All equations must have their variables explained. Currently, it is missing, for example, in Equation 7.

Response 7: Thank you for your comment.We have interpreted the w and b variables in equation 7.

Point 8: Figure 6: labels and text font size is too small. The figure is unreadable.

Response 8: Thank you for your comment. We have enlarged the scale of Figure 8.

Reviewer 2 Report (Previous Reviewer 3)

Please provide a point-to-point response letter based on the reviewers' comments.

This manuscript introduces dimensionality reduction and classification of hyperspectral remote sensing image feature extraction based on machine learning. In summary, the research is interesting and provides valuable results, but the current document has several weaknesses that must be strengthened in order to obtain a documentary result that is equal to the value of the publication.

General considerations:

(1) At the thematic level, the proposal provides a very interesting vision, as the dimensionality reduction and classification of hyperspectral remote sensing image feature extraction is a hot spot in the field of remote sensing research. Nevertheless, a thorough understanding of classification is not limited to the several classification methods used in this paper, which is an important limitation on the desire of the proposal.

(2) The technology proposed in this paper is effective, but the innovation is not prominent. Lack of data support when describing some views.

(3) For the structure of the article, you can combine Chapter 2 and Chapter 3 into one chapter to explain the methodology, so that the structure of the article is better.

Title, Abstract and Keywords:

(4) The abstract is complete and well-structured and explains the contents of the document very well.

Chapter 1: Introduction

(5) The first paragraph introducing the research topic may provide a broader and comprehensive view of the issues related to your topic and you may quote the progress and results of relevant research as much as possible (Wu, F., Duan, J., Ai, P., Chen, Z., Yang, Z., & Zou, X. (2022). Rachis detection and three-dimensional localization of cut off point for vision-based banana robot. Computers and Electronics in Agriculture, 198, 107079. Li, C.; Tang, Y.; Zou, X.; Zhang, P.; Lin, J.; Lian, G.; Pan, Y. A Novel Agricultural Machinery Intelligent Design System Based on Integrating Image Processing and Knowledge Reasoning. Applied Sciences 2022, 12, 7900. ). 

(6) In the field of remote sensing, it is common to use PCA for feature dimensionality reduction and machine learning for classification. Please list the work and results of others in the introduction, and compare them to highlight your innovation.

(7) In general, the research on recognition technology is reasonable, and the interpretation of work objectives may be effective. However, this paper lacks the comparison of the latest methods in this field.

Chapter 2: Hyperspectral Dimensionality Reduction and Classification Methods

(8) In Chapter 2 Hyperspectral Dimensionality Reduction and Classification Methods, please briefly summarize the general methods and highlight the innovation of this article

Chapter 3: Hyperspectral Image Classification Methods

(9) In Chapter 3 Hyperspectral Image Classification Methods, for classification methods, please highlight the innovation of this paper. It is best to put the formula of classification methods to show the theory.

(10) This paper adopts general machine learning methods, but there is too little introduction to neural network. Due to the importance of neural network, it is necessary to increase the elaboration of this content, such as network structure and parameters.

(11) In the third chapter, you may highlight the innovation and focus of this method.

Chapter 4: Experimental Results and Analysis

(12) The last sentence of the second paragraph of this chapter refers to the speed of classification calculation. But there is no data to support this view. Maybe you can include the calculation time and computation in the evaluation index and compare it.

Chapter 5: Conclusion

(13) Conclusion the summary of this paper is comprehensive, and the future work goals can be handled by deep learning methods.

Author Response

General considerations:

Reviewer#3, Concern # 1: At the thematic level, the proposal provides a very interesting vision, as the dimensionality reduction and classification of hyperspectral remote sensing image feature extraction is a hot spot in the field of remote sensing research. Nevertheless, a thorough understanding of classification is not limited to the several classification methods used in this paper, which is an important limitation on the desire of the proposal.

Author response:  Thank you for your comment.

Reviewer#3, Concern # 2: The technology proposed in this paper is effective, but the innovation is not prominent. Lack of data support when describing some views.

Author response:  Thank you for your comment. We added data to the discussion in the paper and obtained hyperspectral downscaled classification data and then performed hypothesis testing to verify the validity of the method.

Author action: We updated the manuscript by including in Yellow in Table 4 ,5

Reviewer#3, Concern # 3: For the structure of the article, you can combine Chapter 2 and Chapter 3 into one chapter to explain the methodology, so that the structure of the article is better.

Author response: Thank you for your comment. The title of the second chapter is modified to hyperspectral dimensionality reduction, and the third chapter is hyperspectral image classification. The analysis of the processing of hyperspectral images from these two directions respectively makes the article more hierarchical.

Author action: We updated the manuscript by including Chapter 2 Hyperspectral Dimensionality Reduction and Chapter 3 Hyperspectral Image Classification Methods.

Title, Abstract and Keywords:

Reviewer#3, Concern # 4: The abstract is complete and well-structured and explains the contents of the document very well.

Author response:  Thank you for your comment.

Chapter 1: Introduction

Reviewer#2, Concern # 5: The first paragraph introducing the research topic may provide a broader and comprehensive view of the issues related to your topic and you may quote the progress and results of relevant research as much as possible.

Author response:  Thank you for your comment. We reviewed a lot more literature and added references related to the topic to summarize the hyperspectral downscaling and classification methods.

Author action: We updated the manuscript by including in Yellow in the first paragraph of introduce.

Reviewer#3, Concern # 6: In the field of remote sensing, it is common to use PCA for feature dimensionality reduction and machine learning for classification. Please list the work and results of others in the introduction, and compare them to highlight your innovation.

Author response:  Thank you for your comment. We compare PCA with the UMAP dimensionality reduction algorithm that works best in our experiments, refer to the deformation of the PCA dimensionality reduction method, and cite its related literature.

Author action: We updated the manuscript by including in Yellow in the penultimate paragraph of 2.4 UMAP.

Reviewer#3, Concern # 7: In general, the research on recognition technology is reasonable, and the interpretation of work objectives may be effective. However, this paper lacks the comparison of the latest methods in this field.

Author response:  Thank you for your comment.The latest method of dimensionality reduction classification of hyperspectral data is deep learning, but deep learning requires a large amount of hyperspectral data and making image labels, and we are sorry that we do not have the ability to complete the experimental comparison. Machine learning can reduce the dimensionality of hyperspectral band data and classify its pixel points, and according to the experimental results, it can also achieve the classification accuracy of deep learning.

Chapter 2: Hyperspectral Dimensionality Reduction

Reviewer#3, Concern # 8: In Chapter 2 Hyperspectral Dimensionality Reduction and Classification Methods, please briefly summarize the general methods and highlight the innovation of this article

Author response: Thank you for your comment. The UMAP algorithm is introduced and compared with the classical PCA dimensionality reduction. Analyzing the advantages of UMAP. The hyperspectral remote sensing images are applied to UMAP algorithm to preserve the global features of the data.

Author action: We updated the manuscript by including in Yellow in the last paragraph of Chapter 2 .

Chapter 3: Hyperspectral Image Classification Methods

Reviewer#3, Concern # 9: In Chapter 3 Hyperspectral Image Classification Methods, for classification methods, please highlight the innovation of this paper. It is best to put the formula of classification methods to show the theory.

Author response: Thank you for your comment. Classification papers for hyperspectral images are divided into two types, one is hard classification, and support vector machine methods can classify the reduced dimensional data well. The other is soft classification, where each pixel is judged as a possible category, which is equivalent to the proportion of various topographies occupied at a single resolution, and neural network classification algorithms can fit the best classification for soft classification of hyperspectral images. We introduce the principles of neural networks and support vector machines and explain the process of classification in remote sensing images.

Author action: We updated the manuscript by including in yellow in Chapter 3.

Reviewer#3, Concern # 10: This paper adopts general machine learning methods, but there is too little introduction to neural network. Due to the importance of neural network, it is necessary to increase the elaboration of this content, such as network structure and parameters.

Author response: Thank you for your comment. We briefly introduced the neural network, we added the structural diagram about the neural network, the number of hidden layer nodes, the selection of activation function, loss function.

Author action: We updated the manuscript by including in yellow in 3.2 Neural Networks.

Reviewer#3, Concern # 11: In the third chapter, you may highlight the innovation and focus of this method.

Author response: Thank you for your comment.  In the face of hyperspectral image classification, both hard classification and soft classification are applied. Hard classification mainly involves deep dimensionality reduction of hyperspectral images first, followed by supervised classification by machine learning. Soft classification is the probability obtained from image classification using logistic regression and neural network, representing the decomposition of hyperspectral image image elements.

Author action: We updated the manuscript by including in yellow in the last paragraph of 3.1 Hard and soft classification for hyperspectral image.

Chapter 4: Experimental Results and Analysis

Reviewer#3, Concern # 12: The last sentence of the second paragraph of this chapter refers to the speed of classification calculation. But there is no data to support this view. Maybe you can include the calculation time and computation in the evaluation index and compare it.

Author response:  Thank you for your comment. We have removed ambiguous statements.

Chapter 5: Conclusion

Reviewer#3, Concern # 13: Conclusion the summary of this paper is comprehensive, and the future work goals can be handled by deep learning methods.

Author response: Thank you for your comment.  We update the goal of our next work, targeting the application of deep learning in hyperspectral image classification for dimensionality reduction.

Author action: We updated the manuscript by including in yellow in Conclusion.

Round 2

Reviewer 1 Report (Previous Reviewer 1)

The  manuscript was well revised and can be accepted for publication.

This manuscript is a resubmission of an earlier submission. The following is a list of the peer review reports and author responses from that submission.

Round 1

Reviewer 1 Report

The paper analyzes dimensionality reduction and machine learning algorithms for dimensionality reduction and classification of remote hyperspectral images. The paper needs to be revised and improved according to the comments and suggestions presented below before it could be considered for publication.

Comments:

1.       Improve the abstract: add a summary of main numerical findings.

2.       Explicitly state your novelty and contribution to the research field.

3.       Figure 1 is trivial. Add more technical details or remove.

4.       The authors discussed various well-known feature dimensionality reduction methods such as PCA, ICA and T-SNE. However, the overview of feature extraction and dimensionality methods in remote sensing research domain is missing. First, start with some review papers which provide a summary of various methods such as in Wambugu, et al. (2021). Hyperspectral image classification on insufficient-sample and feature learning using deep neural networks: A review. Next, analyze specific methodologies presented in various research papers. For example, the authors are encouraged to discuss Khan, et al. (2022). Hyperspectral image classification using NRS with different distance measurement techniques. * Zhou, et al. (2019). Multi-level features extraction for discontinuous target tracking in remote sensing image monitoring. Finally, present a summary of limitations of existing research as a motivation of your study.

5.       The analysis of hyperspectral image classification methods does not have a single supporting references.

6.       The difference between the performance of various feature dimensionality reduction methods presented in Table 4 is very small. Perform statistical analysis of the results. Is the difference statistically significant? Perform statistical testing to confirm or reject the hypothesis of equal means.

7.       Figure 5, 6, 7 are confusing. Since the values at x-axis are independent, use barplots instead of lineplots.

8.       When presenting the coefficient of determination (R^2) values, also present the p-values (Table 5).

9.       Discuss the limitations of your methodology.

10.   Improve conclusions: use main numerical results to support your claims. 

Reviewer 2 Report

This paper analyzed different dimensionality reduction and classification method for hyperspectral image. It is important to compare different DR and FE methods for classification. However,  more conclusions are appreciated for users in their applications of HS classification. And the data used is only one public data, more data or real remote sensed data is also appreciated. The methods compared in this paper are out of date, more recent deep learning methods are missed. 

Reviewer 3 Report

This manuscript introduces dimensionality reduction and classification of hyperspectral remote sensing image feature extraction based on machine learning. In summary, the research is interesting and provides valuable results, but the current document has several weaknesses that must be strengthened in order to obtain a documentary result that is equal to the value of the publication.

General considerations:

(1) At the thematic level, the proposal provides a very interesting vision, as the dimensionality reduction and classification of hyperspectral remote sensing image feature extraction is a hot spot in the field of remote sensing research. Nevertheless, a thorough understanding of classification is not limited to the several classification methods used in this paper, which is an important limitation on the desire of the proposal.

(2) The technology proposed in this paper is effective, but the innovation is not prominent. Lack of data support when describing some views.

(3) For the structure of the article, you can combine Chapter 2 and Chapter 3 into one chapter to explain the methodology, so that the structure of the article is better.

Title, Abstract and Keywords:

(4) The abstract is complete and well-structured and explains the contents of the document very well.

Chapter 1: Introduction

(5) The first paragraph introducing the research topic may provide a broader and comprehensive view of the issues related to your topic and you may quote the progress and results of relevant research as much as possible. 

(6) In the field of remote sensing, it is common to use PCA for feature dimensionality reduction and machine learning for classification. Please list the work and results of others in the introduction, and compare them to highlight your innovation.

(7) In general, the research on recognition technology is reasonable, and the interpretation of work objectives may be effective. However, this paper lacks the comparison of the latest methods in this field.

Chapter 2: Hyperspectral Dimensionality Reduction and Classification Methods

(8) In Chapter 2 Hyperspectral Dimensionality Reduction and Classification Methods, please briefly summarize the general methods and highlight the innovation of this article

Chapter 3: Hyperspectral Image Classification Methods

(9) In Chapter 3 Hyperspectral Image Classification Methods, for classification methods, please highlight the innovation of this paper. It is best to put the formula of classification methods to show the theory.

(10) This paper adopts general machine learning methods, but there is too little introduction to neural network. Due to the importance of neural network, it is necessary to increase the elaboration of this content, such as network structure and parameters.

(11) In the third chapter, you may highlight the innovation and focus of this method.

Chapter 4: Experimental Results and Analysis

(12) The last sentence of the second paragraph of this chapter refers to the speed of classification calculation. But there is no data to support this view. Maybe you can include the calculation time and computation in the evaluation index and compare it.

Chapter 5: Conclusion

(13) Conclusion the summary of this paper is comprehensive, and the future work goals can be handled by deep learning methods.